# Social representation of fair price among professional photographers

Cyrille Feybesse[1,2]*, Todd Lubart[1,2], Leela Rasa[3,4], Charlotte Ossom[3,4], Victor Cavasino[4], Julien Jacob[4], Thibaud Lemonnier[4]

1 LAPEA, Université de Paris, Boulogne-Billancourt, France, 2 Université Gustave Eiffel, Versailles, France, 3 Université de Paris, Boulogne-Billancourt, France, 4 OCUS, Paris, France

* psicy@yahoo.com.br

**Data Availability Statement:** All data are available from the OSF repository (DOI: 10.17605/OSF.IO/KR5ZA).

**Funding:** Funded study This research was financed by Julien Jacob and Thibaud Lemonnier who own the OCS company. URL: https://www.ocus.com/

## Abstract

We investigated the social representation of fair price of French and English-speaking photographers using the free association method. In two independent studies, we performed a factorial analysis of correspondence of the words provided by the participants as well as a similitude analysis. The results indicated that "fair price" was mainly associated with time, effort and experience level of photographers. Both French- and English-speaking samples made similar associations around the concept of fair price but the order of importance varied. We observed some gender-related differences in both samples, although the relative number of male and female participants must be taken into consideration.

## Introduction

In the 21st century's rapidly changing economy there is constant innovation, disruption, and change. The "changing nature of work, flexible work" is cited by 44 percent of business leaders as the top socioeconomic driver of changes in industry [1]. This growing trend is mirrored in worker attitudes with surveys reporting, for example, that 88% of French freelancers do not want to be full-time employees [2]. As attitudes toward the nature of work and technological advances reshape employer-employee relationships, more and more individuals engage in independent work. It is estimated that there are nine and a half million freelancers in Europe and that 20 to 30 percent of the working-age population in the European Union and the United States engage in independent work [2, 3].

Ninety percent of surveyed freelancers reported voluntarily choosing independent work [2]. However, the same respondents also cited difficulty finding clients (65%), identifying income fluctuations (56%), and lack of corporate social benefits (36%) as the frequent concerns faced by these workers. Research has found, for example, that uncertainty is a problem for a majority of television industry workers facing contract changes and that these workers have coped in various ways including diversifying income sources and considering employment in other industries [4]. Other research cites the provision of benefits and income security measures as primary concern across various independent worker profile.

In a world where market mechanisms govern labor outcomes what do workers truly consider to be fair? The concept of a "fair price" is a pervasive one across the social sciences and

en/ They also played a role in the design and in the decision of publishing the results. They did not have had any influence in the analysis, discussion or the discussion of the results. The following co-authors received grant to do this research: CF,TL, LR, CO, VC. The grant number: Erganeo-Q84. Here are the contributions and role played by each author: Ideas, formulation or evolution of overarching research goals and aims: TL, CF, VC, TL and JJ. Data Curation, management activities to annotate (produce metadata), scrub data and maintain research data (including software code, where it is necessary for interpreting the data itself) for initial use and later reuse: CF. Formal Analysis Application of statistical, mathematical, computational, or other formal techniques to analyze or synthesize study data: CF. Funding Acquisition, acquisition of the financial support for the project leading to this publication. TL, JJ. Investigation, conducting a research and investigation process, specifically performing the experiments, or data/evidence collection: LR, CO and VC. Methodology Development or design of methodology; creation of models: CF. Project Administration, management and coordination responsibility for the research activity planning and execution, TL. Resources, provision of study materials, reagents, materials, patients, laboratory samples, animals, instrumentation, computing resources, or other analysis tools: TL and VC. Software, programming, software development; designing computer programs; implementation of the computer code and supporting algorithms; testing of existing code components: CF. Supervision, oversight and leadership responsibility for the research activity planning and execution, including mentorship external to the core team: TL. Validation, verification, whether as a part of the activity or separate, of the overall replication/reproducibility of results/experiments and other research outputs. CYRILLE Visualization Preparation, creation and/or presentation of the published work, specifically visualization/data presentation: CF. Writing – Original Draft Preparation Creation and/or presentation of the published work, specifically writing the initial draft (including substantive translation). CL, CO, LR. Writing – Review & Editing Preparation, creation and/or presentation of the published work by those from the original research group, specifically critical review, commentary or revision – including pre- or post-publication stages. CF, LR, CO, JJ, TL, TL.

**Competing interests:** The authors have declared that no compenting interests exist.

encompasses a wide variety of theories. Despite its importance, the research on this topic has been relatively scarce. The research that does exist has been based off the principle of dual entitlement [5], which argues that firms are entitled to a reference profit whereas customers are entitled to a reference price [6] in their perceptions of fairness. Elsewhere, both the equity theory and the theory of distributive justice suggest that perceptions of fairness occur when a person compares a certain outcome with a comparative other's outcome [7].

It is worth noting that fairness may be conceptually very different from the perception of unfairness; it is possible to be clear about one without having clarity about the other [8]. The concept of unfairness is usually clearer and more concrete than the concept of fairness. People know what is unfair when they see or experience it, but it is difficult to articulate what is fair. Additionally, price evaluations, including fairness assessments, are comparative. Considering these conceptual, cognitive differences, the focus of the present study is on the perception of fairness, without addressing the concept of unfairness. Fairness has previously been defined as "the extent to which outcomes are deemed reasonable and just" ([6] p.475). The cognitive aspect of this definition indicates that price fairness judgments involve a comparison of a price or procedure with a pertinent standard, reference, or norm [7].

With regards to behavioral economics is framed as a social preference for equitable outcomes and an expectation of reciprocity [9]. This is shown in such games as the ultimatum game, where one player is given a sum of money and has to split it with another player, the responder, who may accept it or reject it. If the responder accepts, the money is split per the proposal; if the responder rejects, both players receive nothing. In this game, the proposer has two options: a fair split, or an unfair split. When it comes to fairness and wages, a proposed increase in wages by the employer yields an expectation of reciprocity by the worker with increased effort [10].

The World Fair Trade Organization [11] defines a fair price as "a price that has been mutually accepted during a process of dialogue and consultation, which allows a fair remuneration of producers and is also acceptable by the market in question; [it involves] payment of socially acceptable remuneration considered fair by the producers themselves which takes into consideration the principle of equal pay for men and women with equivalent work". In this respect, the concept and relevant contributing factors of a fair price vary qualitatively across epistemological orientations and are not always easily reconciled. The concept of fair price can in this case be considered as being defined by three components: (1) market price as determined by the equilibrium between supply and demand, (2) the value of labor and (3) the satisfaction of needs [12].

Considering the apparent subjective nature of certain elements such as the satisfaction of needs, the quantification of the contributing factors in determining a fair price remains a process that may fundamentally differ across a variety of properties such as country, culture, gender and even language. As such, the concept of a fair price can be considered as fluid and dynamic in nature. A singular fair price may therefore not exist, such that the concept of a fair price could rather be characterized as one that "fluctuates in space and time [. . .] that are acceptable or less unfair than those of conventional trade" (Diaz, 2007 p.232 cited in [13]). To explore further this issue of fair price, it is necessary to look at social representations.

Social representation refers to the set of knowledge and attitudes about a given object shared by a social group [14]. It is built on the opinions and stereotypes that each member of a group may have towards an object [15] where any given representation will be organized around a nucleus that is relatively stable, with peripheral ideas that are more flexible [16]. In this study, the social representation of "fair price" was explored using a word association task in order to find consensual ideas.

The specific focus of this paper is to explore the representations and ideas that professionals in the photography field have of the concept of fair price. As most work on fair price has focused on basic goods, such as coffee and rice, and to a lesser extent on handicraft goods (cultural craft items), it is important to expand research on fair price to the growing sector of creative activities. Our hope was to provide some indications on what should be considered when one want to reward fairly the service of a photographer. Creative activities represent an important, and growing share of contemporary economies (for example, accounting for 10.2 Billion dollars in the 2015 US economy which is comparable to the transportation sector, according to the Bureau of Economic Analysis). Photography epitomizes the cultural activity sector, as photography is structured around independent producers (freelance in particular) who are confronted with an unregulated marketplace for their work.

A word association study was conducted with two independent samples, in order to determine the main concepts associated with fair price. Photographers work worldwide but French-and English-speaking photographers are two representative groups which allow general social representations and possible cultural variations to be explored. We decided to use this method because it allows us to understand how fair price is understood and the subjects associated with it. The social representation method is ideal to explore this topic.

We were interested in addressing two main questions concerning the concept of fair price when used in a context of creative productions: 1) is the concept of fair price relatively the same across different professional photographers? 2) does the social representation of fair price vary by gender or culture?

## Material and methods

### Method

All the participants were contacted by email and participated in an online Questionnaire about fair price. They all had to provide informed consent before assessing the web questionnaire and all the data were analysed anonymously. In France, ethics committees are used for biomedical research with measures, such as blood or saliva, drug treatment, evoked potential and other specific biological measures and all research using at-risk populations (such as pregnant women, patients with illness, etc.). Based on the national research center's recommendations, studies involving questionnaires with healthy populations are not subject to specific ethics committee procedures. The study presented in the article is an internet-based questionnaire on social concept of fair price, applied to a population with no specific risk factors. The study considers the social representation of what a "fair price" is to professional photographers who contribute to OCUS (an internet-based service proposing short term jobs to independent photographers) was analyzed. In this context a professional photographer is "someone qualified to accept paid photography jobs". This was done using a word-association method where participants wrote the first six words that came to their mind when reading the words "fair price".

The word analysis could only consider a single language, so all participants who did not provide words written in French or in English were excluded. This allowed analysis of French words from a reasonably heterogeneous sample of participants located in France and English words from a more culturally and geographically diverse participant sample. The final sample consisted of 767 out of the 784 participants recruited. The data was analysed with two independent groups: English-speaking participants (N = 471) and French-speaking participants (N = 296).

Before performing the different analyses, the English and French words were separately transformed: verbs were put into the infinitive, nouns and adjectives into the singular and similar expression such as "fairness" or "single words like "fair" for example. The second step of

the analysis was to proceed with a correspondence factor analysis of the words (Reinert Method) provided in English and French. This analysis examines the associations generated for the inductive word. The factorial analysis in social representation studies is a qualitative and exploratory method that is helpful to group and summarize the words of a given database. The words provided by the participants are grouped according to their repeated association (different participants providing the same couple of words). The P value allow us to check the degree of relationship between the word and its class (group). Finally, the factor analysis plot allows us to observe the relationship between the different classes. Based on participants' language (English or French) and gender (male or female), differences in the social representation of fair price were examined to see if they provided different matrices of similarities. The results provided a representational graphic with the central theme and the expressions associated with it, as well as secondary themes and the words associated with those.

## French-speakers' social representation of fair price

### Participants

The French-speaking sample had 296 participants (221 males and 74 females) with an average age of 41.45 years (SD = 11.76). The participants came mainly from France (97.6%), with others located in Belgium and Brazil. Most of the participants declared themselves as self-employed or independent freelancers (89.5%) with the remaining sample declaring themselves as having either part-time or full-time employee status (8.5%).

### Results

The French-speaking sample provided 2426 words in which 479 (19%) words appeared only once and were not included in the analysis because they are considered idiosyncratic, not clearly part of a socially-shared representation. Additionally, 160 words were each written by at least three different participants and these are the words used in the analysis (see Table 1). There were a considerable number of words cited at least 50 times by the participants: temps (time), équitable (fair), travail (work), prix (price) and qualité (quality). A further 15 words were cited between 15 and 47 times, and are shown in Table 1. Some of these words are respect (respect), remuneration (remunération), honnêteté (honesty) or valeur (value) for example. These words could be defining the concept of fair price. The first result indicates a rather homogeneous set of ideas and representations about fair price. The word cloud generated by the frequency of words is presented in Fig 1. It indicates the concepts associated with fair price. The primary associations made with the concept of a fair price were: time, quality, work and respect.

### Correspondence factor analysis

The correspondence analysis considered the words that were cited at least three times by participants (see Table 2). The analysis indicated a solution with 5 classes forming two distinct groups and 80% of the segments could be classified which denotes a good quality analysis. The first group was formed by classes 1, 3 and 4 that are very interdependent. This first group included 62.5% of the words in the analysis (see Fig 2). This means that the same words were strongly associated in two or three of these classes (see Fig 2). The chi calculation shows no significant difference between these three classes. The second group was formed by factors 2 and 5 and these two classes are independent of each other. This second group of classes included 37.5% of the words in the factor analysis. This suggests that the main words associated to each class (2 and 5) are not relevant to the other classes.

**Table 1. Word frequencies found with the French-speaking sample.**

| Word | Frequency | Word | Frequency |
|------|-----------|------|-----------|
| Temps | 66 | Rémunérateur | 8 |
| Equitable | 65 | Raisonnable | 8 |
| Travail | 59 | Payer | 8 |
| Prix | 56 | Marge | 8 |
| Qualité | 50 | Equilibre | 8 |
| Respect | 47 | Deplacement | 8 |
| rémunération | 35 | Argent | 8 |
| Honnète | 32 | Adapter | 8 |
| Valeur | 28 | Viable | 7 |
| Correct | 27 | Valorisation | 7 |
| Matériel | 26 | Talent | 7 |
| Charge | 26 | Remuneration | 7 |
| reconnaissance | 25 | Professionnalisme | 7 |
| Vivre | 22 | Prestation | 7 |
| Salaire | 21 | Mission | 7 |
| investissement | 21 | Heure | 7 |
| Frais | 21 | Equité | 7 |
| déplacement | 21 | Equilibre | 7 |
| Rentable | 20 | Distance | 7 |
| professionnel | 20 | Confort | 7 |
| Normal | 16 | Conforme | 7 |
| Marché | 16 | Compte | 7 |
| Compétence | 16 | Cohérent | 7 |
| Valorisant | 14 | Motivation | 6 |
| Suffisant | 14 | Honnète | 6 |
| Rentabilité | 14 | Bénéfice | 6 |
| Rapport | 13 | Adéquation | 6 |
| Expérience | 13 | Sérieux | 6 |
| Respectueux | 11 | Technique | 5 |
| Justice | 11 | savoir faire | 5 |
| Tarif | 10 | Remunération | 5 |
| Satisfaction | 10 | Qualite | 5 |
| Mériter | 10 | Permettre | 5 |
| Gagnant | 10 | Légitime | 5 |
| Prendre | 9 | Gain | 5 |
| Passer | 9 | Echange | 5 |
| Motivant | 9 | Décent | 5 |
| Minimum | 9 | Competence | 5 |
| Justifier | 9 | Benefice | 5 |
| Horaire | 9 | Auteur | 5 |
| Experience | 9 | Win | 4 |
| Droit | 9 | Taxe | 4 |
| Confiance | 9 | Service | 4 |
| Client | 9 | Revenir | 4 |
| amortissement | 9 | Réalité | 4 |
| Acceptable | 9 | Précis | 3 |
| Vie | 8 | Plaisir | 3 |
| Satisfaisant | 8 | Partager | 3 |

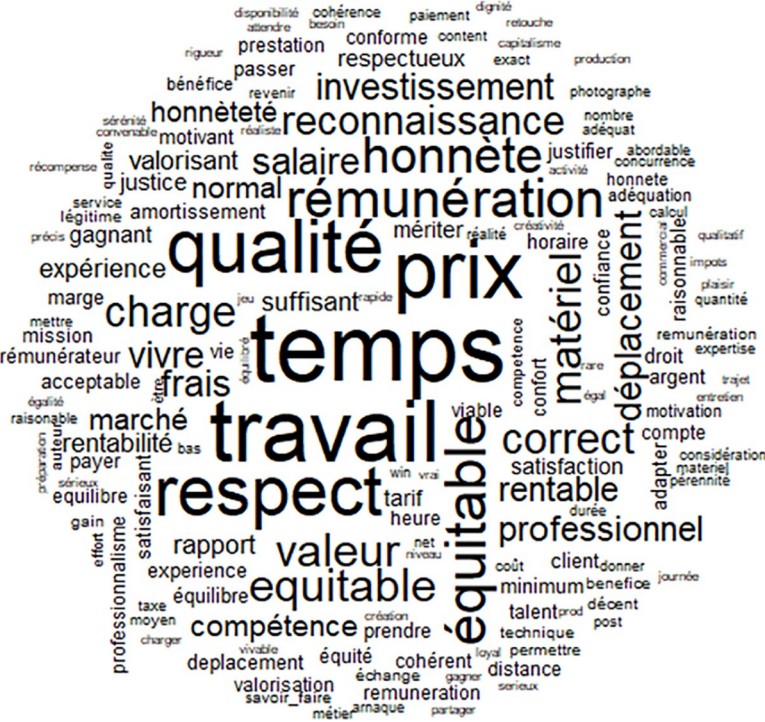

**Fig 1. Word cloud provided by the male and female French-speaking participants.**

Class 5 is the most independent class. Participants who cited the words in class 5 tended to not cite words present in the other classes (see Fig 2). The themes in classes 1, 3 and 4 seem to be relatively similar and closely related. It was observed that some words like "competence", "equity", "valorisation" and "motivation" were present in more than one class. Some words like "confidence" and "loyal" or "justice" and "merit" were synonymous to each other. These classes lean towards what a fair price should be and explore the relationship between two parties (employer and employee). The employer should be just and able to recognize the quality of the work of the employee, rewarding according to his or her qualities.

The second group, composed of classes 2 and 5, focuses on the practical aspects of price. The top words for both classes addressed some prerequisites for doing a job (travel, margin, adaptive, time or amortization). The main idea here is that a fair price must account for the costs of a job.

## Similitude analysis

A similitude analysis was conducted, comparing the social representation of women and men. Weaker loadings were found with the female participants because there were many more male participants than female participants. For both groups, fair price was associated with positive concrete and abstract attributes. Female participants seemed to have focused on fewer aspects than the male participants did. It was expected, however, to find fewer clusters in this sample because we have fewer data from female participants.

The female sample had two strong clusters (see Fig 3). The words gravitated around time and respect which would suggest that these two concepts are central to women. Time had more word associations and was related to words like "being paid", "material" or "investment". This suggests that "time" tends to be related to the practical aspects of a job. Respect was

**Table 2. Classes obtained with the words provided by the French-speaking participants.**

| P | >0.0001 |
|---|---|
| | >0.0001 |
| | 0.0068 |
| | 0.019 |
| | 0.019 |
| | 0.019 |
| | 0.023 |
| **Class 5** | Marge |
| | Permettre |
| | Rapport |
| | Amortissem |
| | Compte |
| | Revenir |
| | Prendre |
| P | 0.00134 |
| | 0.0017 |
| | 0.006 |
| | 0.0105 |
| | 0.014 |
| | 0.0149 |
| | 0.0149 |
| | 0.032 |
| **Class 4** | Frais |
| | Motivant |
| | Raisonable |
| | Expérience |
| | Valorisation |
| | Minimum |
| | Vie |
| | Compétence |
| P | >0.00001 |
| | 0.00026 |
| | 0.003 |
| | 0.004 |
| | 0.02 |
| | 0.036 |
| | 0.036 |
| | 0.035 |
| **Class 3** | Justice |
| | Valorisant |
| | Gagnant |
| | Equitable |
| | Motivation |
| | Heure |
| | Raisonable |
| | Loyal |
| | Normal |
| P | >0.0001 |
| | 0.003 |
| | 0.0032 |
| | 0.0032 |
| | 0.034 |
| | 0.007 |
| | 0.02 |
| | 0.03 |
| | 0.035 |
| | 0.046 |
| | 0.046 |

*(Continued)*

**Table 2.** (Continued)

| Class 2 | Déplacement |
| | Qualité |
| | Rémunerateu |
| | Adapter |
| | Légitime |
| | Matériel |
| | Equitable |
| | Temps |
| | Trajet |
| | Prendre |
| | Expérience |
| P | 0.004 |
| | 0.010 |
| | 0.020 |
| | 0.020 |
| | 0.020 |
| | 0.023 |
| | 0.026 |
| | 0.030 |
| | 0.040 |
| Class 1 | Valeur Compétence |
| | Confiance |
| | Client |
| | Expérience |
| | Equité |
| | Rémunération |
| | Mérité |
| | Temps |

Note: P indicates the significance level of association between the word and the class (Chi-square)

associated with words like "equity", "acceptable", "work" and "professionalism" so fair price is something that respects the person being paid.

The male similitude analysis provided more data and clusters (see Fig 4). The central thematic of male participants was work. This central word was surrounded by three other clusters formed with the words "time", "respect" and "price". The relationship between work and fair price was associated with words such as "trust", "salary", "profession", which are qualities that a given work should have and are related to both the employer and the employee. For all the other clusters, the idea of equity, and fairness remained present. For "respect" for example, the same idea was observed among women that fair price must consider the quality and competence. Fair price was related to taking into account the quality, time spent to do a job and price.

## English-speakers' social representation of fair price

### Participants

The English-speaking sample was larger and more heterogeneous than the French one. There were 471 participants from 44 different countries, with the largest number of participants coming from India (14%) followed by the UK (12.7%), Spain (8.7%), Portugal (7%) and Colombia (7%). As found with the French sample, most of the participants were male (80%) and the mean average for the whole sample was 37.16 years (SD = 11.52). In this sample, 89.5% were self-employed, 8.5% were employed full or part-time and the rest were either students, unemployed or retired.

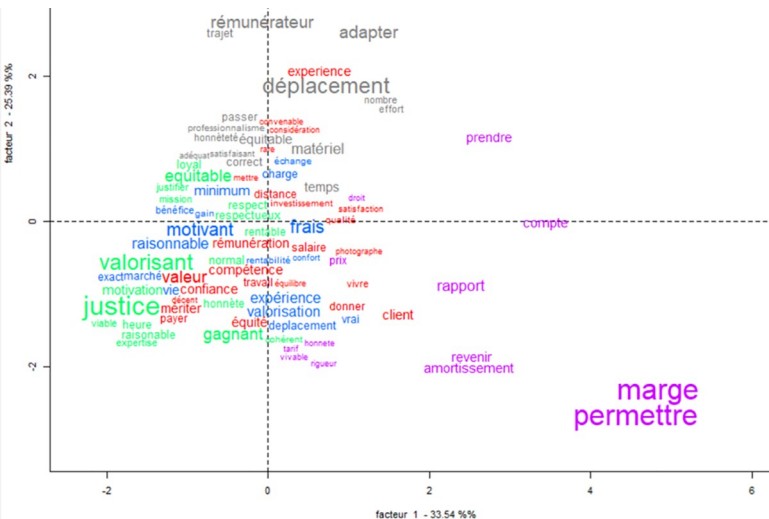

**Fig 2. Factor analysis plot for the words associated to fair price among male and female French-speaking participants.**

## Results

This sample provided a pool of 4260 words in which 809 (19%) were cited only once and were excluded from the final analysis. A total of 227 words were written by at least by three different participants from the sample and were considered in this analysis. This indicates that participants had a wide array of ideas and showed more heterogeneity than the French sample. This

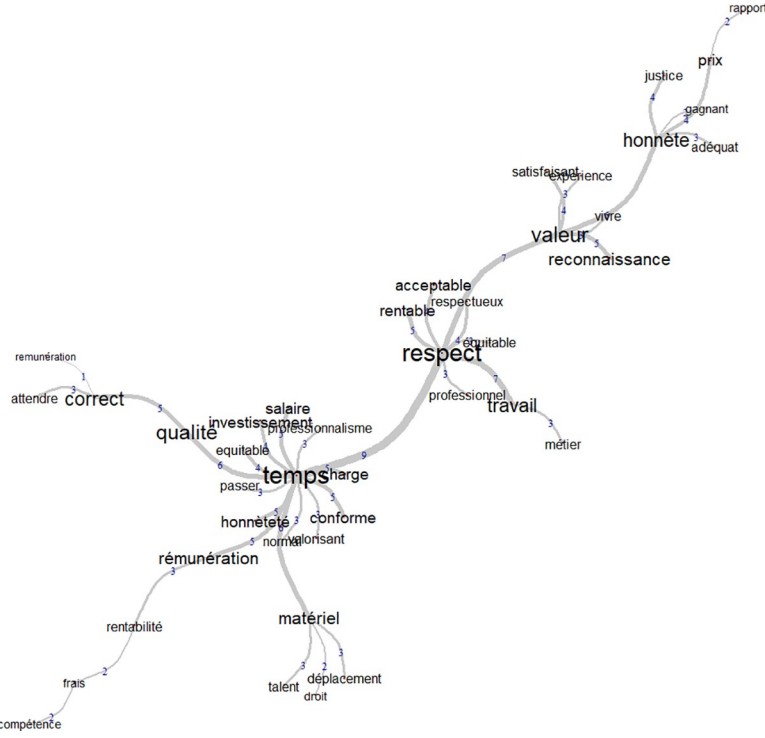

**Fig 3. Word similitudes among French-speaking women.**

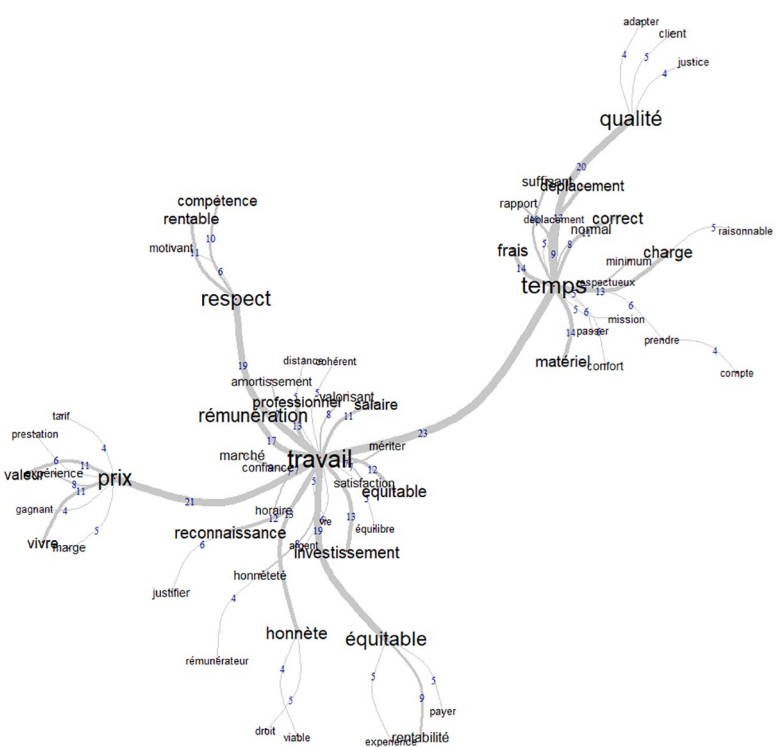

**Fig 4. Word similitudes among French-speaking men.**

could be due to the fact that the sample is composed of people from multiple countries around the world, whereas the French sample was primarily composed of people from France.

The frequency of words revealed that 8 words were cited more than 50 times and 18 words were cited between 20 and 49 times (see Table 3). They seem to be addressing similar themes to those cited by the French-speaking sample. The most frequent words were also highlighted in the word cloud generated (Fig 5). The concept of fair price was mostly associated with price (98), work (92), good (86), fair (78), time (72), value (69), reasonable (65) and quality (58).

## Correspondence factor analysis

The factor analysis generated 5 classes and 85.32% of the segments could be classified. In the current case, class 2 is mixed with class 5 and class 4 (Fig 6). This means that the words cited in class 2 were associated with words cited in classes 5 and 4. As seen in Fig 6, these three classes are very related and included 57.3% of the words included in the analysis. Classes 1 and 5 are more independent so the words associated with the central words of these classes were different from the words of the other classes. Table 4 shows the words most associated for each class. There are noticeably more words associated with classes 1 and 3 than with class 2, and 5. This latter set of classes seemed to highlight the qualities such as hard work or skill that should be associated with fair price. These words relate to receiving a fair amount of money for doing a photography job (photo, photographer, equipment, expenses. . .). Class 1 seemed to be oriented towards the profession of photography with the association of fair price using words such as professionalism, hour, edit, decent, competition, and proper. Finally, class 3 could be related to outcomes associated with a fair price (happiness, professional, unbiased, deserve, legal, honest. . .).

**Table 3. Word frequencies found with the English-speaking sample.**

| Word | Frequence | Word | Frequency |
|---|---|---|---|
| Price | 98 | satistaction | 13 |
| Work | 92 | Budget | 13 |
| Good | 86 | Live | 12 |
| Fair | 78 | Standard | 12 |
| Time | 72 | Rate | 12 |
| Value | 69 | motivation | 12 |
| Reasonable | 65 | recognition | 12 |
| Quality | 58 | Base | 11 |
| Honest | 43 | Knowledge | 11 |
| Money | 36 | Adequate | 11 |
| Pay | 36 | Interest | 11 |
| Market | 30 | Fairness | 10 |
| Right | 29 | Amount | 10 |
| Reward | 27 | professionalism | 10 |
| compensation | 26 | opportunity | 10 |
| Happy | 26 | Client | 10 |
| Equipment | 25 | Happiness | 10 |
| Correct | 25 | Fee | 9 |
| Expense | 24 | Acceptable | 9 |
| Effort | 23 | Minimum | 9 |
| Job | 23 | Creativity | 9 |
| Respect | 23 | Sufficient | 9 |
| Professional | 22 | Skill | 9 |
| Experience | 20 | Day | 9 |
| Profit | 20 | Worthy | 9 |
| Justice | 19 | consideration | 8 |
| Balance | 19 | Study | 8 |
| Competitive | 19 | Payment | 8 |
| Cost | 18 | Cheap | 8 |
| Honesty | 18 | photography | 8 |
| Equality | 17 | Trust | 8 |
| Low | 17 | Decent | 8 |
| Worth | 16 | Life | 8 |
| Deal | 16 | Profitable | 8 |
| Equitable | 16 | Income | 8 |
| Satisfy | 15 | Charge | 7 |
| Win | 15 | Respectful | 7 |
| Sustainable | 15 | Business | 7 |
| Equal | 15 | Agreement | 7 |
| Hour | 14 | Customer | 7 |
| Average | 14 | comfortable | 7 |
| Deserve | 14 | Cover | 7 |
| Level | 13 | Person | 7 |
| Travel | 13 | Benefit | 7 |
| Affordable | 13 | Include | 6 |

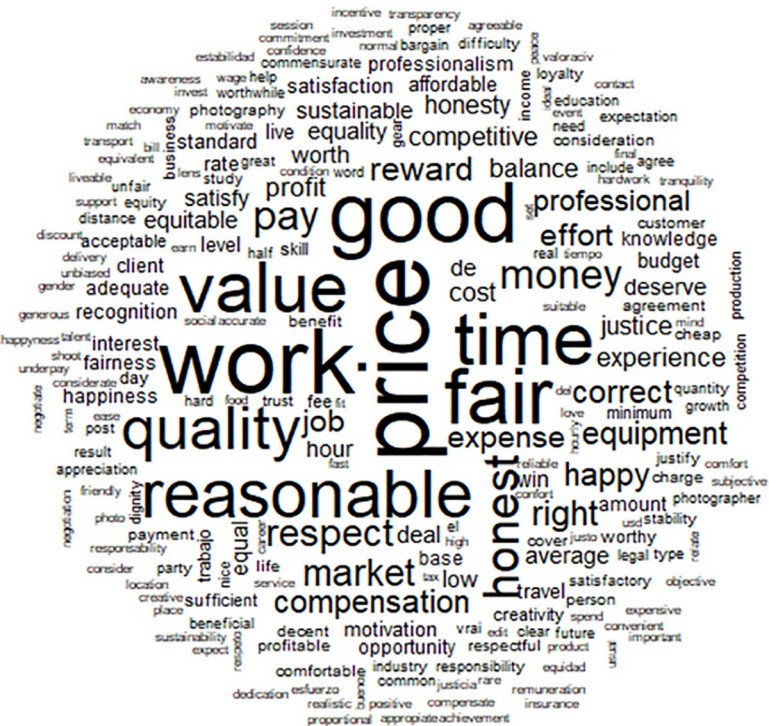

**Fig 5. Words cloud provided by the male and female English-speaking participants.**

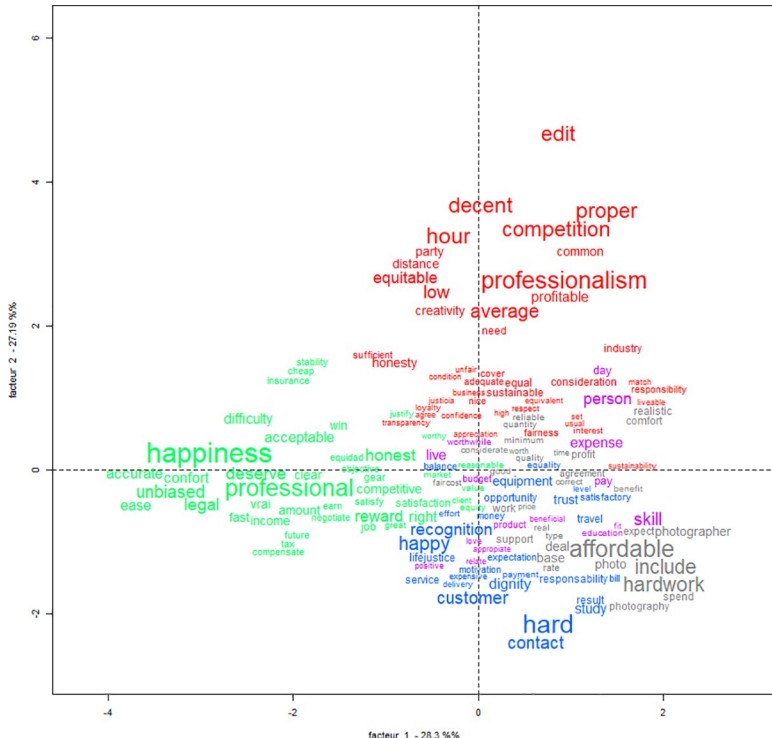

**Fig 6. Factor analysis plot for the words associated to fair price among male and female English-speaking participants.**

**Table 4. Classes obtained with the words provided by the English-speaking participants.**

| | |
|---|---|
| P | 0.00035<br>0.00173<br>0.00253<br>0.00337<br>0.02122<br>0.02244 |
| **Class 5** | Skill<br>Person Expenses Live<br>Day Pay |
| P | <0.0001<br>0.00011<br>0.0004<br>0.00051<br>0.00068<br>0.00288<br>0.00470<br>0.00741<br>0.02630<br>0.02480<br>0.03917<br>0.03917<br>0.03920 |
| **Class 4** | Hard<br>Happy Contact Customer Recognition Dignity Study Equipment Trust<br>Life Result Service<br>Responsability |
| P | <0.0001<br><0.0001<br>0.00014<br>0.00063<br>0.00066<br>0.00075<br>0.00154<br>0.0024<br>0.0024<br>0.0024<br>0.0029<br>0.00323<br>0.00362<br>0.00677<br>0.0152<br>0.0189 |
| **Class 3** | Happiness<br>Professional Unbiased Deserve Legal Honest Reward Comfort Accurate Ease<br>Right Acceptable Difficulty Competitive Win<br>Amount |
| P | <0.0001<br><0.0001<br><0.0001<br>0.00708<br>0.00708<br>0.00708<br>0.00708 |
| **Class 2** | Affordable<br>Include Hard work Deal<br>Base Photographer Photo |

(*Continued*)

**Table 4.** (Continued)

| P | |
|---|---|
| | <0.0001 |
| | <0.0001 |
| | <0.0001 |
| | <0.0001 |
| | <0.0001 |
| | <0.0001 |
| | 0.00016 |
| | 0.00055 |
| | 0.00181 |
| | 0.00547 |
| | 0.00814 |
| | 0.01456 |
| | 0.01456 |
| | 0.01456 |
| | 0.01786 |
| | 0.02616 |
| **Class 1** | Professionalism |
| | Hour Edit Decent |
| | Competition Proper Average |
| | Low Equitable Profitable Honesty Party Distance Common Creativity |
| | Equal |

Note: P indicates the significance level of association between the word and the class (Chi-square)

## Similitude analysis

Similar to the French-speaking female group, there were fewer and weaker clusters in the English-speaking female group because there were fewer female participants. In this case, the central word is "good" and the words "value", "reasonable", "fair" and "work" gravitated around it. The words "fair" and "value" have better clusters than the other main words (see Fig 7). The central word "good" was surrounded by words such as profession, compensation, respect or payment. This could be seen as a reference to positive attributes towards fair price. The word "fair" was mainly associated with effort, time and experience. The word "honest" had less significance compared to the female French data and had more positive attributes attached to it with words like "satisfying", "happy" and "recognition".

The male group had very clear clusters. "Time" was the central word and can be considered to be the main concept for fair price (see Fig 8). This suggests the amount of time one gives to a specific work should be the best way to determine a fair price. The word "time" is related to the words "good", "value", "fair", "price" and "quality". These can be seen as other important aspects of fair price. Fair was mainly associated with synonyms of the idea of fairness and is also attached to the word "reasonable". This might be just indicating different ways of speaking about what a fair price is, as something that is just or correct. The word "quality" was associated with the word "work" so a fair price would require work done with quality. The word "good" seems to be related to personal consequences of fair price (see Fig 8).

The similitude analysis suggests that there is a similar view of fair price across gender. It is hard to determine the role of culture in this case because there are answers from several countries, but the English-speaking sample appears to differ slightly from what was found in the French-speaking sample.

## Discussion

This social representation analysis of a fair price revealed that some common themes associated with the concept of fair price. In both studies, the idea of time and work are considered as

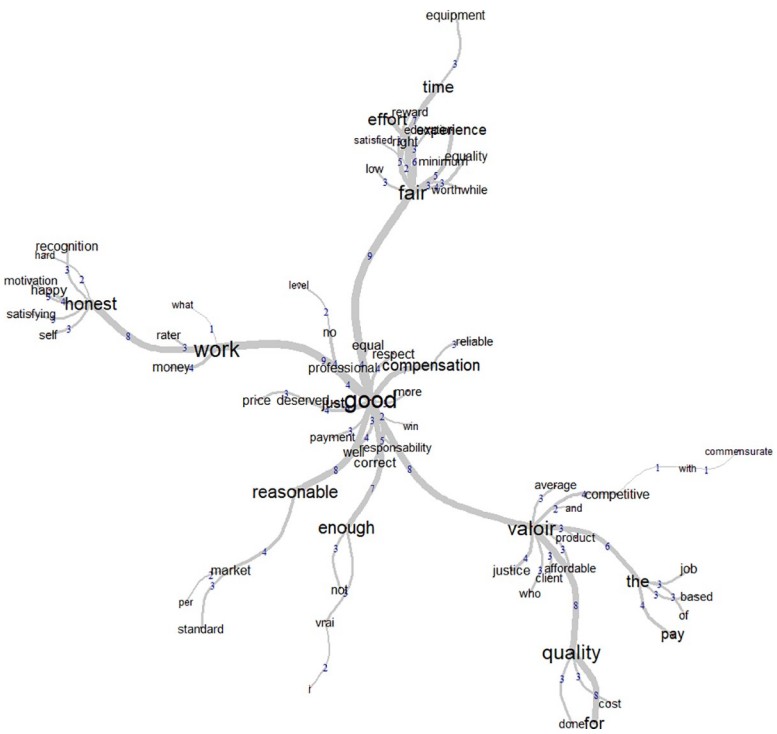

**Fig 7. Word similitudes among English-speaking women.**

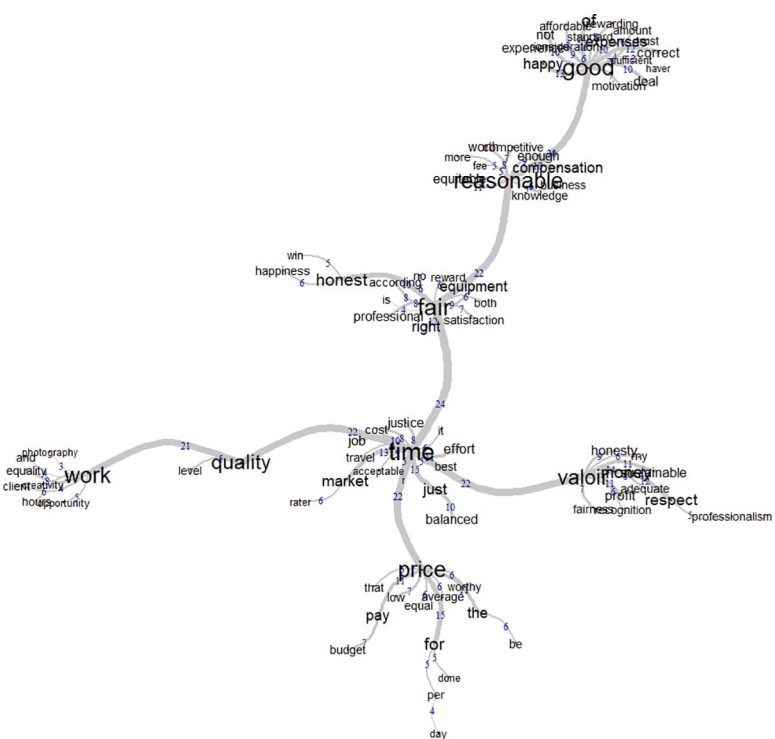

**Fig 8. Word similitudes among English-speaking men.**

central to fair price. There was also an observed association of difficulty and quality around the idea of work. Another word that was highlighted was "professionalism" which suggests that the professional level of the photographer should be considered when determining how much a given job should be paid. A secondary and less strongly associated aspect to fair price was the technical aspect of a job–related to transportation, equipment, and effort. A third aspect associated with fair price was the idea of justice, equity or fairness. This suggests that receiving an acceptable compensation for a given job leads the participants to experience these types of feelings.

This same indication was found with several positive attributes associated with the concept of fair price such as happiness, reward, or the idea of winning. The French-speaking sample focused more on the practical aspects that are required to do a professional photography job. The English-speaking sample seemed to be focused on what is needed; the concepts of competition and hard work seem to be more present in this sample when compared to the French sample.

We can conclude that the participants of our sample gravitated around the same ideas about fair price although the factorial analysis of words indicated similar but different classes between the French and English-speaking participants. This reveals that there are some cross-cultural aspects not on how fair price is conceived but rather on what is important in this concept.

The similitude analysis of the different samples showed several clusters with similar central words. Time, work and value were associated with the concept of fair price. For both English-speaking and French-speaking samples, fair price was associated with positive attributes. Gender differences were examined in each sample and the central words for men and women were found to be different in both cases. This could indicate that the gender of the participant impacts the aspects that are related to fair price. Given the uneven number of male and female participants in both samples, strong conclusions cannot be drawn and further research is needed.

## Conclusion

The results suggest that the concept of fair price is complex and multifaceted. It appears to vary cross-culturally and according to gender. Of the professional photographers registered to create content with OCUS, the French-speaking sample was relatively homogeneous compared to the English-speaking one, which was composed of people from all over the world. Despite the industry specific nature of the sample frequently cited words like fair, price, quality, and value are consistent with broader economic conceptions of fair price [14]. The central word in the similitude analysis was not the same for the two samples and might indicate that although different cultures agree on what is important to take into account to establish what a fair price is, the different cultural and linguistic groups differ on the level of importance given to these clusters. These results suggest that companies that hire third parties for handicraft artwork should know what aspects the locals consider to be more important so they can agree on what is the fair price for the service. We believe that these conclusions can be also useful for other professionals related to the field of handicraft artwork.

Further research into cultural variation in fair price should consider national social policy factors (i.e. national health care, unemployment benefits, and retirement savings) that may impact an individual's expectations from an employment exchange due to national variation in coverage and access based on employment status. Follow up research could explore the representation of an unfair price and the quantification of these conceptual representations as research suggests that fairness judgments involve a comparison to a reference point [12]. As

the boundaries and definition of employer-employee relationships change, companies and platforms that empower workers through collaboratively developing ways to ensure income equity and security will help to mitigate some of the primary challenges of independent workers.

## Author Contributions

**Conceptualization:** Cyrille Feybesse, Todd Lubart, Charlotte Ossom.

**Data curation:** Cyrille Feybesse, Charlotte Ossom.

**Formal analysis:** Cyrille Feybesse, Leela Rasa.

**Funding acquisition:** Julien Jacob, Thibaud Lemonnier.

**Investigation:** Cyrille Feybesse, Todd Lubart, Leela Rasa.

**Methodology:** Cyrille Feybesse.

**Project administration:** Todd Lubart, Charlotte Ossom, Victor Cavasino, Julien Jacob, Thibaud Lemonnier.

**Resources:** Julien Jacob, Thibaud Lemonnier.

**Software:** Cyrille Feybesse.

**Supervision:** Todd Lubart.

**Validation:** Cyrille Feybesse.

**Visualization:** Todd Lubart, Victor Cavasino, Julien Jacob.

**Writing – original draft:** Cyrille Feybesse, Leela Rasa, Charlotte Ossom.

**Writing – review & editing:** Cyrille Feybesse, Todd Lubart, Leela Rasa, Charlotte Ossom.

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
