## [Decision Letter · Decision Letter 0]

8 Jul 2020

PONE-D-20-12797

Social representation of fair price among professional photographers

PLOS ONE

Dear Dr. Feybesse,

Thank you for submitting your manuscript to PLOS ONE. After careful consideration, we feel that it has merit but does not fully meet PLOS ONE’s publication criteria as it currently stands. Therefore, we invite you to submit a revised version of the manuscript that addresses the points raised during the review process.

Both qualified reviewers liked your paper but also raised questions for improvement. I have read your paper carefully and agree with the reviewers. Please try to address both reviewers’ concerns as much as you can in the revision.

Reviewer 1’s comments and suggestions are more demanding. I do not suggest you re-do the survey but if you can that would be great. To address reviewer 1’s concerns, you need to motivate your research better and more clearly and also tone town your research goal. Reviewer 2 suggests the introduction be shortened which I totally agree.

I also suggest you provide more explanation on the correspondence factor analysis method, as reviewer 1 suggests. For Figures 3, 4, 7, 8, is it possible to use different colors or add numbers to indicate class 1-5? I find it hard to match the classes in the figure with classes in Tables 2 and 4 because I need to look for specific words. Also can you add a title for each figure indicating whether the figure is for male or female? The current figures 7 and 8 are uploaded in reverse order so it is easy to get confused without a figure title.

Typo: Line 277, Figure 8 should be Figure 6?

We look forward to receiving your revised manuscript.

Kind regards,

Shihe Fu, Ph.D.

Academic Editor

PLOS ONE

Journal Requirements:

3. Thank you for including your funding statement; "

Funded study

This research was financed by Julien Jacob and Thibaud Lemonnier who own the OCS company. URL: https://www.ocus.com/en/   They also played a role in the design and in the decision of publishing the results. They did not have had any influence in the analysis, discussion or the discussion of the results.

The following co-authors received grant to do this research: CF,TL, LR, CO, VC. The grant number: Erganeo-Q84.

Here are the contributions and role played by each author:

Ideas, formulation or evolution of overarching research goals and aims:  TL, CF, VC, TL and JJ.

Data Curation, management activities to annotate (produce metadata), scrub data and maintain research data (including software code, where it is necessary for interpreting the data itself) for initial use and later reuse: CF.

Formal Analysis    Application of statistical, mathematical, computational, or other formal techniques to analyze or synthesize study data: CF.

Funding Acquisition, acquisition of the financial support for the project leading to this publication.   TL, JJ.

Investigation, conducting a research and investigation process, specifically performing the experiments, or data/evidence collection: LR, CO and VC.      

Methodology

Development or design of methodology; creation of models: CF.

Project Administration,    management and coordination responsibility for the research activity planning and execution, TL.

Resources, provision of study materials, reagents, materials, patients, laboratory samples, animals, instrumentation, computing resources, or other analysis tools: TL and VC.  

Software, programming, software development; designing computer programs; implementation of the computer code and supporting algorithms; testing of existing code components: CF.

Supervision, oversight and leadership responsibility for the research activity planning and execution, including mentorship external to the core team: TL.

Validation, verification, whether as a part of the activity or separate, of the overall replication/reproducibility of results/experiments and other research outputs.  CYRILLE

Visualization    Preparation, creation and/or presentation of the published work, specifically visualization/data presentation:  CF.

Writing – Original Draft Preparation    Creation and/or presentation of the published work, specifically writing the initial draft (including substantive translation).  CL, CO, LR.

Writing – Review & Editing    Preparation, creation and/or presentation of the published work by those from the original research group, specifically critical review, commentary or revision – including pre- or post-publication stages.  CF, LR, CO, JJ, TL, TL]

3.

We note that you have indicated that data from this study are available upon request. PLOS only allows data to be available upon request if there are legal or ethical restrictions on sharing data publicly. For more information on unacceptable data access restrictions, please see http://journals.plos.org/plosone/s/data-availability#loc-unacceptable-data-access-restrictions.

Reviewers' comments:

Reviewer's Responses to Questions

**Comments to the Author**

1. Is the manuscript technically sound, and do the data support the conclusions?

Reviewer #1: Partly

Reviewer #2: Yes

2. Has the statistical analysis been performed appropriately and rigorously? 

Reviewer #1: Yes

Reviewer #2: Yes

3. Have the authors made all data underlying the findings in their manuscript fully available?

Reviewer #1: Yes

Reviewer #2: Yes

4. Is the manuscript presented in an intelligible fashion and written in standard English?

Reviewer #1: Yes

Reviewer #2: Yes

5. Review Comments to the Author

Reviewer #1: Summary of the research

This study concerns an important question of what is a fair price for handicraft artworks. Using a word-association method, the authors surveyed samples of French-speaking and English-speaking professional photographers to get a sense of what are related to “fair price”. A few more analyses were done to help identify the associations between the reported words as well as gender differences. Then conclusion was drawn that the most common theme of the social representations of the concept of fair price are time and work (related to the production of the photos), despite of differences across French and English cultures and between genders.

Overall Impression

The research question is interesting and important. The research method is straightforward and efficient in answering your research question. However, I recommend a major revision for this manuscript. First it remains questionable whether your method is optimal for your research question. Second it seems to me that you have just touched the surface of this important question of how people think of the fair prices of handicraft artworks, without digging deeper into it. Let me elaborate below by highlighting two major issues.

Major Issues 1

The first major issue relates to the design of your method. You need to at least explain why you chose this method and why this particular type of survey question is used. The current survey question is extremely simple and efficient in getting a quick view of how photographers think about fair price, but perhaps it is too simple to draw valuable conclusions from the data. To be more precise, the main issue here is the ambiguity of these words. It may be natural, for example, to think of time spent on the production of works when you see the reported word “time”, but how can you be sure that is exactly what the respondents mean? How can you know “time” does not mean, e.g. the fair price of a photo work can only be revealed over a period of time? And similarly, how do you know that “work” indicates the amount of work needed for production rather than the (value) of the piece of work to be traded? For many words reported by the respondents, there is the problem of ambiguity. So the general concern to me is that you are just telling one of many possible stories here. Although from the association study the readers can more or less infer what the respondents could mean by these words, you can never be sure of it.

This problem could be eased through various ways. For example, you can allow them to use phrases rather than single words only, to make them express themselves more precisely. Or you may design more concrete questions, putting them into specific scenarios and let them choose among options. Anyway, the key suggestion here is that you should at least try to reduce the ambiguity by modifying your survey questions or add questions, or something else.

Major Issue 2

The second major issue relates to the depth of your research question. It seems to me you could have explored deeper into what people think determine the fair price of handicraft artworks. Think about your contribution (suppose the first issue mention above is perfectly solved): the paper would let readers know that professional photographers think fair prices are those that match the time and work devoted into their works. To me this finding is neither surprising nor complete. Do you think this result is representative at the society level? In other words, do you think other groups of people (e.g. consumers of photography works) would have similar views of “fair price” as professional photographers? And do you think your finding has implications with respect to how governments should regulate the pricing of handicraft artworks?

In general, you have touched an important research question, but you have restricted your sample to producers of photography works, and as a result, you can only claim that your finding indicates what professional photographers think, rather than what the society think about the fair price of craft photo works. What really matters, however, is the latter. You can focus on photography industry. This is absolutely fine. But if I were you I would survey more groups of people, especially the consumers of those photographers, or even those who have nothing to do with photography. The key suggestion here is that you extend your sample from producers of photo works to consumers and other social groups of people, surveying about their view of fair prices of photo works. It seems to me the specific question you are currently investigating is of lower significance than what the journal expects.

To sum up, regarding the major issues, I’d like to see two changes: a modified survey to reduce word ambiguity, and a bigger sample including more social groups. In addition, there are some minor issues which I think you need to address.

Minor Issues

1. I didn’t see English literature relating to “social representation” and to the concept of “fair price”. (At least you have mentioned the research of fair prices of basic goods. Where is the corresponding literature?) You should add your review of these literatures.

2. Does your survey include demographic questions such as income level and work experience as photographers? If yes, you should analysis these data; if not you should care about these data as these factors might affect people’s view of fair prices.

3. I think you should explain the correspondence factor analyses more clearly with respect to how to understand Figure 2 and Figure 6. I felt confused about these figures and cannot understand them fully by reading your corresponding texts.

Overall, for the reasons explained above, I recommend a major revision.

Reviewer #2: Review of “Social representation of fair price among professional photographers” PONE-D-20-12797

.

Summary:

This paper investigates the social representation of fair price of French and English-speaking photographers using the see association method. They perform a factorial analysis of correspondence of the words provided by the participants as well as a similitude analysis via an online questionnaire.

Results:

What they found is that “fair price” was mainly associated with time, effort and experience level of photographers. Culture matters but not much.

Evaluation:

I like the topic, price plays a central role in economics and fairness is much studied in behavioral economics, however, what determines fair price is still understudied in economics. This paper provides a descriptive way to find out which concepts are determined for the feeling of fairness of price. I have some comments listed below:

1) In my view, this paper seems to be a bit too simple, this does not mean simplicity is not appreciated, the authors use a questionnaire to elicit people’s opinion about fair price, however, the authors do not discuss fully the mechanism, just simply put the findings as it is, is there a “correlation” between fair price and words? Or does just the frequency of words matter?

2) I think the authors should get right to the point, say, research question. I find the first three paragraphs are unnecessarily long, the authors can shorten them, or move some of them to the footnote.

3) How the authors choose questionnaire participants? Are they representative?

4) I find the discussion part is not enough, the authors use two studies and just find there are some small differences? Why?

5) Policy implication part should also be discussed a bit more.

6) About fair price, the authors can borrow ideas from some behavioral economists, like Richard Thaler for example, to discuss a bit what behinds these words and why this is associated with fair price.

6. PLOS authors have the option to publish the peer review history of their article (what does this mean?). If published, this will include your full peer review and any attached files.

Reviewer #1: No

Reviewer #2: No

---

## [Author Response · Author response to Decision Letter 0]

1 Oct 2020

Dear Editors and reviewers of PLOS ONE,

First of all, thank you all for taking the time to review our manuscript. I found all the comments were very good and really help to improve the article. Also, your feedbacks were provided with very good guidance.

I sincerely tried to address the points that were raised but I am willing to work on them again if you ever think that this new version is not good enough. We were able to reduce the intro like everybody have suggested. 

I would like to address some points that were raised by both reviewers.

Reviewer #1

I agree that it would be interesting to have different participants that provide handicraft artworks but this research was financed by a photography company which allow us to get access to a lot of photographers. They were mainly concerned in promoting a better relationship among the photographers that work with them. As a researcher, I was interested in exploring the concept of fair price with their sample and provide some practical guidance to them. This is something I could do when I presented the results. 

In the new version of the manuscript, I tried to address why the method of social representation is an interesting method for this kind of research. It allowed us to explore freely how fair price is conceived on people that have to deal with it in their professional life. 

I agree about the ambiguity this type of studies provide. It is hard to be completely sure about what these words are saying. This couId be dealt with if we had other social representation studies on the topic of fair price or among photographers. I did not find any. I preferred, and still prefer, in staying more pragmatic about what we can say so we avoid misleading the readers. 

Regarding the minor issues you have addressed, I did not have more demographics questions than the ones we presented in the manuscript. I tried to provide more explanations about the factorial analysis method like you have suggested. 

Reviewer #2

I really appreciated your comments and I have to say that I am new in this world of fairness and economy. I’m not sure that I understood what you meant by “correlation between fair price and words”. What a study using the social representation method does is to ask people to provide words that they associate with the inductive word. The frequency of a word is very important here because it reveals a common association between two words. 

I formulated the research question as you have requested. Our main concern was to see how the photographers see fair price. The sample we had available allow us to explore this in a cross-cultural perspective. 

It is hard to say if the sample we’ve got is representative but I believe it could be considered to be when we are dealing with photographers. I found it hard to generalise our results to other kinds of handicraft artwork but I think this work might be helpful. 

 Thank you, once again, for all your great comments. I hope you all will be satisfied with this new version. We are open to new suggestions. 

With my best regards,

Dr. CF

---

## [Decision Letter · Decision Letter 1]

24 Nov 2020

Social representation of fair price among professional photographers

PONE-D-20-12797R1

Dear Dr. Feybesse,

We’re pleased to inform you that your manuscript has been judged scientifically suitable for publication and will be formally accepted for publication once it meets all outstanding technical requirements.

Kind regards,

Shihe Fu, Ph.D.

Academic Editor

PLOS ONE

Additional Editor Comments (optional):

Reviewers' comments:

Reviewer's Responses to Questions

**Comments to the Author**

1. If the authors have adequately addressed your comments raised in a previous round of review and you feel that this manuscript is now acceptable for publication, you may indicate that here to bypass the “Comments to the Author” section, enter your conflict of interest statement in the “Confidential to Editor” section, and submit your "Accept" recommendation.

Reviewer #1: All comments have been addressed

Reviewer #2: All comments have been addressed

2. Is the manuscript technically sound, and do the data support the conclusions?

Reviewer #1: (No Response)

Reviewer #2: Yes

3. Has the statistical analysis been performed appropriately and rigorously? 

Reviewer #1: (No Response)

Reviewer #2: Yes

4. Have the authors made all data underlying the findings in their manuscript fully available?

Reviewer #1: (No Response)

Reviewer #2: Yes

5. Is the manuscript presented in an intelligible fashion and written in standard English?

Reviewer #1: (No Response)

Reviewer #2: Yes

6. Review Comments to the Author

Reviewer #1: From a pragmatic point of view, the paper addressed your research question well, and that's why I decided to suggest acceptance, although a deeper question could have been analyzed if the sample were extended and the ambiguity problem were dealt with. I encourage you to carry on further study with the topic of social representation of "fair price", and try to relate your study to theories (either economic theories or sociological theories) that provide guidance to the understanding of the concept of fairness or fair price.

Reviewer #2: (No Response)

7. PLOS authors have the option to publish the peer review history of their article (what does this mean?). If published, this will include your full peer review and any attached files.

Reviewer #1: No

Reviewer #2: No

---

## [Editor Report · Acceptance letter]

26 Nov 2020

PONE-D-20-12797R1 

Social Representation of Fair Price among Professional Photographers 

Dear Dr. Feybesse:

I'm pleased to inform you that your manuscript has been deemed suitable for publication in PLOS ONE. Congratulations! Your manuscript is now with our production department. 

Kind regards, 

on behalf of

Dr. Shihe Fu 

Academic Editor

PLOS ONE